# Interval-Censored Regression with Non-Proportional Hazards with Applications

**Fábio Prataviera** [1,†] , **Elizabeth M. Hashimoto** [2,†] , **Edwin M. M. Ortega** [1,*,†] , **Taciana V. Savian** [1,†]
**and Gauss M. Cordeiro** [3,†]

1   Department of Exact Sciences, "Luiz de Queiroz" School of Agriculture, University of São Paulo—ESALQ/USP, Piracicaba 13418-900, Brazil; fabio_prataviera@usp.br (F.P.); tvsavian@usp.br (T.V.S.)
2   Academic Department of Mathematics, Federal University of Technology, Londrina 86036-370, Brazil; emiehashimoto@gmail.com
3   Department of Statistics, Federal University of Pernambuco, Recife 50670-901, Brazil; gausscordeiro@gmail.com
*   Correspondence: edwin@usp.br
†   These authors contributed equally to this work.

**Abstract:** Proportional hazards models and, in some situations, accelerated failure time models, are not suitable for analyzing data when the failure ratio between two individuals is not constant. We present a Weibull accelerated failure time model with covariables on the location and scale parameters. By considering the effects of covariables not only on the location parameter, but also on the scale, a regression should be able to adequately describe the difference between treatments. In addition, the deviance residuals adapted for data with the interval censored and the exact time of failure proved to be satisfactory to verify the fit of the model. This information favors the Weibull regression as an alternative to the proportional hazards models without masking the effect of the explanatory variables.

**Keywords:** interval-censored data; non-proportional hazards; regression model; residual deviance; survival analysis





## 1. Introduction

Data that represent the time to event are characterized by the presence of censorship. In other words, they are partial observations of the response variable caused by the non-occurrence of the event of interest during the period under study. Interval-censored data arise when the occurrence time of the event is only known to have occurred in an interval. Interval censoring is a type of censorship that occurs naturally in studies in which the sample unit is evaluated periodically [1]. Such data appear, for example, in clinical or longitudinal experiments in which patients can be followed up only through periodic examinations. In this case, we do not know the exact occurrence time, but only that it occurs within an interval. Recently, some works on interval-censored data have been reported. In Betensky et al. [2], the authors introduced proportional hazards regressions, Calle and Gómez [3] used a Bayesian framework for analyzing regressions in which one of the covariables is interval-censored, Komárek and Lesaffre [4] proposed an accelerated failure time model, and Rodrigues et al. [5] discussed an alternative to the Kaplan–Meier method.

However, these models [2,6–9] are defined under the proportional hazards assumptions, i.e., the survival curves exhibit parallel behavior. However, these assumptions do not always hold.

In fact, what can be said is that most studies adopt the assumption of proportional risks, i.e., the survival curves do not intersect. For example, Cox's model considers this assumption. In this research, the risks may be non-proportional.

On the other hand, in a study of dairy cow herds developed by the Department of Veterinary Medicine of the Federal University of Lavras, MG, in Brazil, to evaluate the effect of animal supplementation on the ovulation of dairy cows (interval-censored survival data), the survival curve of the control supplementation was crossed with the survival curve of the new supplementation (see Figure 1a). Another example refers to breast cancer data, where the goal is to compare the effects of radiotherapy alone versus radiotherapy and adjuvant chemotherapy on women. In Figure 1b, we present the graph of the survival curve with the two effects.

In both cases, proportional hazards models are not suitable for analyzing ovulation times and compare radiotherapy and radiotherapy + chemotherapy treatments, and as an alternative, we can consider accelerated failure time models. However, in some situations, the accelerated failure time models may not be able to capture the difference between treatments, possibly due to interference from the crossing of survival curves, as shown in Hashimoto et al. [1]. A solution to solve the effect of crossed survival curves (non-proportional hazards assumption) is to consider the scale parameter as a systematic component of the model.

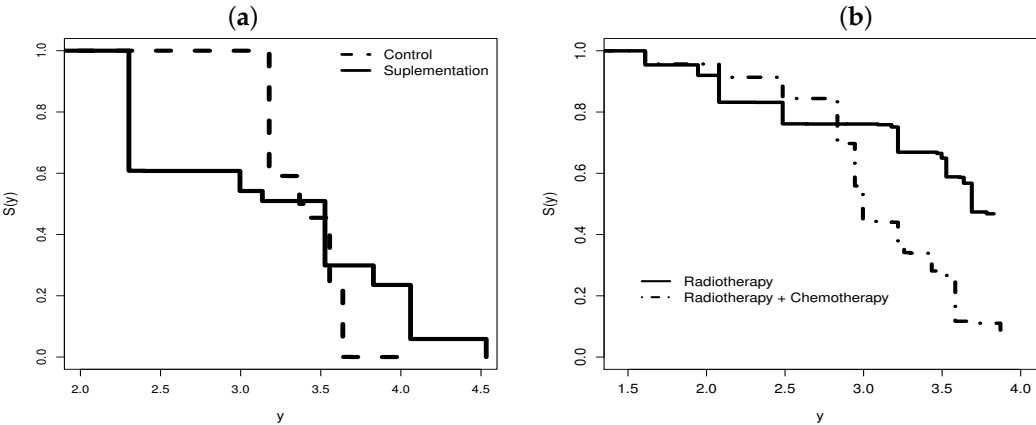

**Figure 1.** Estimated survival curve by Turbull method: (**a**) For control and new supplementation. (**b**) By radiotherapy and radiotherapy + chemotherapy treatments.

In this context, we include the effects of covariables in the location and scale parameters in the Weibull accelerated model with two systematic components to analyze interval-censored data with non-proportional hazards. We use standard likelihood theory for inferential purposes. Another objective is to propose an extension of the deviance residuals [1] for interval-censored survival data without considering the proportional hazards assumption.

The paper is organized as follows. In Section 2, we propose the log-Weibull regression with two systematic components for the location and scale parameters for interval-censored data. The estimation of the parameters by maximum likelihood is addressed in Section 3. We provide a simulation study to check the accuracy of the estimates and residuals in Section 4. In Section 5, we empirically prove the utility of the new regression. We offer some concluding remarks in Section 6.

## 2. The Proposed Model

The Weibull and log-Weibull distributions are frequently adopted for analyzing censored data [10–13] and phenomena with monotone failure rates [14]. Let $T \sim W(\alpha, \lambda)$ be the Weibull random variable, where $\alpha > 0$ is the shape parameter and $\lambda > 0$ is the scale parameter. The probability density function (pdf) of the log-Weibull defined by $Y = \log(T)$

under the re-parametrization $\sigma = \alpha^{-1} > 0$ (scale) and $\mu = \log(\lambda) \in \mathbb{R}$ (location), say $Y \sim \text{LW}(\mu, \sigma)$, has the form

$$f(y; \mu, \sigma) = \frac{1}{\sigma} \exp\left[\left(\frac{y - \mu}{\sigma}\right) - \exp\left(\frac{y - \mu}{\sigma}\right)\right], \quad y \in \mathbb{R}. \tag{1}$$

The log-exponential distribution comes with $\sigma = 1$.

The survival function corresponding to (1) is

$$S(y; \mu, \sigma) = \exp\left[-\exp\left(\frac{y - \mu}{\sigma}\right)\right]. \tag{2}$$

Clearly, the random variable $Z = (Y - \mu)/\sigma$ has density

$$f(z) = \exp[z - \exp(z)], \quad z \in \mathbb{R}. \tag{3}$$

The accelerated failure time parametric model, also known as the location-scale regression [14], has been used for studying the effects of covariables on the response variable. Let $Y_1, \cdots, Y_n$ be independent random variables such that $Y_i \sim \text{LW}(\mu_i, \sigma_i)$. In addition, we consider the vectors of explanatory variables $\mathbf{x}_i = (1, x_{i1}, \cdots, x_{ip})^\top$ and $\mathbf{w}_i = (1, w_{i1}, \cdots, w_{iq})^\top$. The linear location–scale regression for interval-censored data is defined as by (for $i = 1, \ldots, n$)

$$Y_i = \mu_i + \sigma_i Z_i, \tag{4}$$

where $\mu_i = \mathbf{x}_i^\top \boldsymbol{\beta}_1$ and $\log(\sigma_i) = \mathbf{w}_i^\top \boldsymbol{\beta}_2$, the random error $Z_i$ has density function (3), and $\boldsymbol{\beta}_1 = (\beta_{10}, \cdots, \beta_{1p})^\top$ and $\boldsymbol{\beta}_2 = (\beta_{20}, \cdots, \beta_{2q})^\top$ are functionally independent.

Thus, we associate a systematic component with the parameter $\sigma$ to allow us to model data with non-proportional risks and the presence of heteroscedasticity. Equation (4) can be useful for presenting such data.

For interval-censored data, the logarithms of the failure times $Y_i$ (for $i = 1, \ldots, n$) are not observed exactly, only that the event occurred at some moment in a interval time of the type $(\log(U_i), \log(V_i)]$, where $\log(U_i) < Y_i < \log(V_i)$. An exact failure time is observed if $\log(U_i) = \log(V_i)$. In practice, the values of $\log(U_i)$ and $\log(V_i)$ refer to evaluation times, and hence for some individuals the event can occur after the last visit to the evaluator, thus characterizing observations subject to right censoring, where the failure time $Y_i$ occurs in the interval $[\log(U_i), \infty)$. A summary of the structure of the data is given below:

- Exact failure time $\rightarrow Y_i = y_i$.
- Right-censored $\rightarrow Y_i \in [\log(U_i), \infty)$.
- Interval-censored $\rightarrow Y_i \in (\log(U_i), \log(V_i)]$.

## 3. Estimation

We are only interested in models with an exact failure time and interval censoring. Let $(\log(u_1), \log(v_1), \mathbf{x}_1, \mathbf{w}_1), \cdots (\log(u_n), \log(v_n), \mathbf{x}_n, \mathbf{w}_n)$ be a set of interval-censored observations. The log-likelihood function of the log-Weibull regression for an exact failure time and interval-censored data has the form

$$\begin{aligned}
l(\boldsymbol{\theta}) = \sum_{i \in E} &\left\{-\mathbf{w}_i^\top \boldsymbol{\beta}_2 + \left[\frac{\log(u_i) - \mathbf{x}_i^\top \boldsymbol{\beta}_1}{\exp(\mathbf{w}_i^\top \boldsymbol{\beta}_2)}\right] - \exp\left[\frac{\log(u_i) - \mathbf{x}_i^\top \boldsymbol{\beta}_1}{\exp(\mathbf{w}_i^\top \boldsymbol{\beta}_2)}\right]\right\} - \\
&\sum_{i \in R} \exp\left\{\frac{\log(u_i) - \mathbf{x}_i^\top \boldsymbol{\beta}_1}{\exp(\mathbf{w}_i^\top \boldsymbol{\beta}_2)}\right\} + \\
&\sum_{i \in I} \log\left\{\exp\left[-\exp\left(\frac{\log(u_i) - \mathbf{x}_i^\top \boldsymbol{\beta}_1}{\exp(\mathbf{w}_i^\top \boldsymbol{\beta}_2)}\right)\right] - \exp\left[-\exp\left(\frac{\log(v_i) - \mathbf{x}_i^\top \boldsymbol{\beta}_1}{\exp(\mathbf{w}_i^\top \boldsymbol{\beta}_2)}\right)\right]\right\},
\end{aligned} \tag{5}$$

where $\boldsymbol{\theta} = (\boldsymbol{\beta}_1^\top, \boldsymbol{\beta}_2^\top)^\top$ is the vector of parameters, and $E$, $R$, and $I$ denote sets of elements with exact failure time, right censoring and interval censoring, respectively.

The maximum likelihood estimate (MLE) $\hat{\boldsymbol{\theta}}$ of $\boldsymbol{\theta}$ can be found by maximizing (5) in numerical platforms such as `Ox 8.00` (MaxBFGS function) or `R` Version 3.6.3.

Inference can be based on the $(p + q + 2) \times (p + q + 2)$ observed information matrix $\mathbf{J} = \mathbf{J}(\boldsymbol{\theta})$. The asymptotic distribution of $(\hat{\boldsymbol{\theta}} - \boldsymbol{\theta})$ can be considered $N_{p+q+2}(0, \mathbf{J}(\boldsymbol{\theta})^{-1})$ under first-order asymptotes. We construct confidence intervals for the parameters based on the approximate multivariate normal $N_{p+q+2}(0, \mathbf{J}(\hat{\boldsymbol{\theta}})^{-1})$ distribution of $\hat{\boldsymbol{\theta}}$.

## 4. Modified Deviance Residuals

Various types of residuals have been proposed in the regression literature [14]. For some applications of censored data, see, for example, the log-Birnbaum–Saunders regression [15] and the generalized log-gamma regression [16]. Furthermore, [1] presented martingale residuals for interval-censored data. Following these ideas, we extend these residuals with no proportional risk assumption.

The observed sample $\{[\log(u_i), \log(v_i)], i = 1, \ldots, n\}$ is taken from Section 3. The martingale residuals [1,17,18] for interval-censored data have the form

$$
r_{M_i} = \begin{cases}
1 + \log[\hat{S}(\log(u_i))] & \text{if} \quad i \in E, \\[2mm]
\log[\hat{S}(\log(u_i))] & \text{if} \quad i \in R, \\[2mm]
\dfrac{\hat{S}(\log(u_i)) \log[\hat{S}(\log(u_i))] - \hat{S}(\log(v_i)) \log[\hat{S}(\log(v_i))]}{\hat{S}(\log(u_i)) - \hat{S}(\log(v_i))} & \text{if} \quad i \in I,
\end{cases}
$$

where $\hat{S}(\cdot)$ is the estimated survival function. So, replacing $\hat{S}(\cdot)$ in Equation (2), we obtain the martingale residuals for the log-Weibull regression for interval-censored data and non-proportional hazards:

$$
r_{M_i} = \begin{cases}
1 - \exp(z_{ui}) & \text{if} \quad i \in E, \\[2mm]
-\exp(z_{ui}) & \text{if} \quad i \in R, \\[2mm]
\dfrac{\exp[zv_i - \exp(zv_i)] - \exp[zu_i - \exp(zu_i)]}{\exp[-\exp(zu_i)] - \exp[-\exp(zv_i)]} & \text{if} \quad i \in I,
\end{cases}
\tag{6}
$$

where

$$
zu_i = \frac{\log(u_i) - \mathbf{x}_i^\top \hat{\boldsymbol{\beta}}_1}{\exp(\mathbf{w}_i^\top \hat{\boldsymbol{\beta}}_2)} \quad \text{and} \quad zv_i = \frac{\log(v_i) - \mathbf{x}_i^\top \hat{\boldsymbol{\beta}}_1}{\exp(\mathbf{w}_i^\top \hat{\boldsymbol{\beta}}_2)}.
$$

However, the martingale residuals are asymmetrical and have values less than or equal to one. In contrast, [1] used an adaptation of the deviance residuals [17] for interval-censored data. Based on a combination of these residuals, we define the *m*odified deviance residuals by

$$
r_{D_i} = \begin{cases}
\text{sgn}(r_{M_i}) \left[-2\{r_{M_i} + \log(1 - r_{M_i})\}\right]^{1/2} & \text{if} \quad i \in E, I \\[2mm]
\text{sgn}(r_{M_i}) \left(-2 r_{M_i}\right)^{1/2} & \text{if} \quad i \in R,
\end{cases}
\tag{7}
$$

where $r_{M_i}$ are the martingale residuals given by (6).

*Simulation Studies*

1. **Deviance residual**

A first simulation study with 1000 replications analyzed the empirical distribution of the residuals (7) by combining $n = 30, 50, 100$ and $300$ with the interval censoring

percentages 100%, 90%, and 70%. For each combination, the generated data follow the proposal by [1]:

(a)   The coefficients are fixed at $\beta_{10} = 3.00$, $\beta_{11} = 0.72$, $\beta_{20} = -0.71$ and $\beta_{21} = 0.60$;
(b)   The explanatory variable $x_1$ is generated from the binomial distribution with a success probability equal to 0.5 and 1, and $\mu_i = \beta_{10} + \beta_{11} x_1$ and $\sigma_i = \exp(\beta_{20} + \beta_{21} x_1)$ are computed;
(c)   The variable $z = \log[-\log(1-u)]$ is generated from $u \sim U(0,1)$, and the logarithm time from (4);
(d)   The logarithm censoring time is generated from a uniform distribution $[0, \tau]$ by fixing $\tau$ until right-censoring percentages of 0, 10%, or 30%;
(e)   The interval length is generated from a uniform $U(1,3)$ distribution;
(f)   The limits of the log interval time are $li = y + (1+k)/2$ and $ls = y - (1+k)/2$, where $k$ is the length of the randomly chosen interval.

For each fit, we calculate the residuals $r_{D_i}$ and construct the QQ plots displayed in Figure 2. Table 1 reports the averages of the MLEs of the parameters and mean square errors (MSEs) of the Weibull regression, respectively.

**Table 1.** Averages of the MLEs and MSE (in parentheses) for the parameters of the Weibull regression model for interval-censored data.

| $n$ | Parameters | Actual Values | 0% | 10% | 30% |
|-----|-----------|--------------|-----|------|------|
| 30 | $\beta_{10}$ | 3.00 | 2.9158 (0.0284) | 2.9313 (0.0279) | 2.9354 (0.0352) |
|  | $\beta_{11}$ | 0.72 | 0.7444 (0.0879) | 0.7286 (0.0965) | 0.7275 (0.1245) |
|  | $\beta_{20}$ | −0.71 | −1.1733 (0.8250) | −1.2731 (1.3701) | −1.4154 (2.2010) |
|  | $\beta_{21}$ | 0.60 | 0.8623 (0.7849) | 0.9578 (1.3332) | 1.0628 (2.2225) |
| 50 | $\beta_{10}$ | 3.00 | 2.9260 (0.0114) | 2.9304 (0.0132) | 2.9411 (0.0172) |
|  | $\beta_{11}$ | 0.72 | 0.7416 (0.0456) | 0.7427 (0.0573) | 0.7375 (0.0714) |
|  | $\beta_{20}$ | −0.71 | −1.0239 (0.1120) | −1.0507 (0.2259) | −1.0612 (0.3456) |
|  | $\beta_{21}$ | 0.60 | 0.7701 (0.1679) | 0.7848 (0.2642) | 0.7950 (0.3919) |
| 100 | $\beta_{10}$ | 3.00 | 2.9401 (0.0055) | 2.9443 (0.00964) | 2.9571 (0.0086) |
|  | $\beta_{11}$ | 0.72 | 0.7408 (0.0216) | 0.7372 (0.0275) | 0.7302 (0.0353) |
|  | $\beta_{20}$ | −0.71 | −0.9714 (0.0390) | −0.9680 (0.0382) | −0.9461 (0.0429) |
|  | $\beta_{21}$ | 0.60 | 0.7502 (0.0554) | 0.7479 (0.0597) | 0.7319 (0.0678) |
| 300 | $\beta_{10}$ | 3.00 | 2.9437 (0.0018) | 2.9468 (0.0022) | 2.9603 (0.0028) |
|  | $\beta_{11}$ | 0.72 | 0.7438 (0.0075) | 0.7436 (0.0086) | 0.7398 (0.0117) |
|  | $\beta_{20}$ | −0.71 | −0.9381 (0.0098) | −0.9263 (0.0102) | −0.8997 (0.0112) |
|  | $\beta_{21}$ | 0.60 | 0.7335 (0.0156) | 0.7257 (0.0170) | 0.7101 (0.0204) |

The numbers in Table 1 indicate that the MSEs of the estimates decay toward zero when $n$ increases. However, for the estimate of $\sigma$, although its MSE is decreasing, the estimated bias does not change, which may be influenced by the presence of interval censoring. We can note in Figure 2, regardless of $n$, that there is a greater displacement of the residuals when the percentage of interval censorship decreases. This is because the sample is being contaminated by interval censoring.

## 2. Estimating the survival function

We construct a second simulation study with 200 replicates to estimate the survival functions with covariables on the parameters $\mu$ and $\sigma$ simultaneously and on the parameter $\mu$.

For the simulation scenarios, we set the sample size $n = 100$ combined with the same censoring percentages of the previous study. The data are also generated in the same way given in the item *a*. However, using the same generated data, the estimation is conducted as follows:

(a)   Effects of the covariables on the parameters $\mu$ and $\sigma$ simultaneously:

i    Initial values $\beta_{10} = 3.00$, $\beta_{11} = 0.72$, $\beta_{20} = -0.71$ and $\beta_{21} = 0.60$;

ii   Estimate the survival function by

$$\hat{S}(y_i) = \exp\left\{ -\exp\left[ \frac{y_i - (\hat{\beta}_{10} + \hat{\beta}_{11} x_{1i})}{\exp(\hat{\beta}_{20} + \hat{\beta}_{21} x_{1i})} \right] \right\}, \quad i = 1, \dots 100,$$

where $x_1$ is generated from a binomial distribution with a success probability 0.5 and 1.

(b)   Effects of the covariables on the parameter $\mu$:

i    Initial values $\beta_{10} = 3.00$, $\beta_{11} = 0.72$, and $\sigma = 0.6129$, where the initial value for $\sigma$ is the estimate given in Table 2;

ii   Estimate the survival function by

$$\hat{S}(y_i) = \exp\left\{ -\exp\left[ \frac{y_i - (\hat{\beta}_{10} + \hat{\beta}_{11} x_{1i})}{\hat{\sigma}} \right] \right\}, \quad i = 1, \dots 100,$$

where $x_1$ is generated from a binomial distribution with a success probability equal to 0.5 and 1.

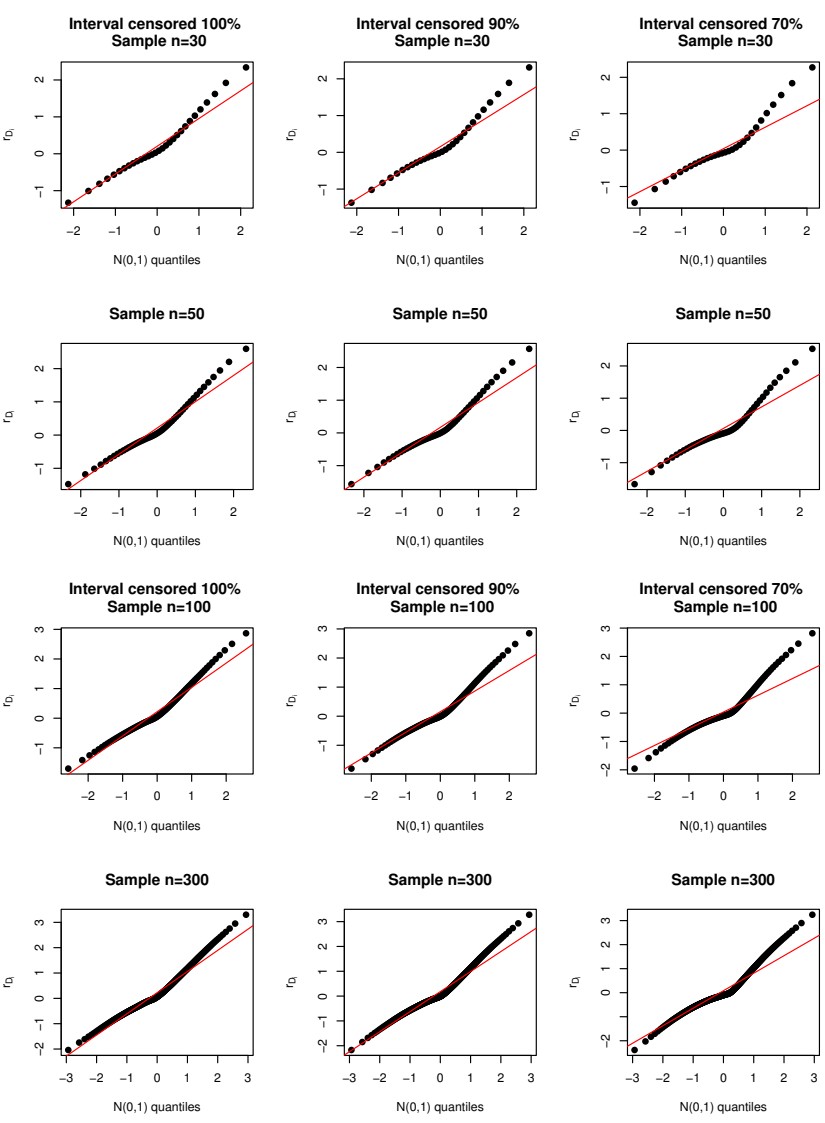

**Figure 2.** QQ plots for the residuals.

**Table 2.** Some findings from the fitted regressions.

| | One Component for $\mu$ | | | | Two Components for $\mu$ and $\sigma$ | | |
|---|---|---|---|---|---|---|---|
| Parameter | Estimate | SE | *p*-Value | B | Estimate | SE | *p*-Value |
| $\beta_{10}$ | 3.3311 | 0.1053 | <0.0001 | $\beta_{10}$ | 3.3329 | 0.0852 | <0.0001 |
| $\beta_{11}$ | 0.5648 | 0.1738 | 0.0016 | $\beta_{11}$ | 0.7193 | 0.2418 | 0.0037 |
| $\sigma$ | 0.6129 | 0.0721 | - | $\beta_{20}$ | −0.7088 | 0.1407 | <0.0001 |
| | | | | $\beta_{21}$ | 0.5952 | 0.2558 | 0.0221 |
| $-2l(\boldsymbol{\theta}) = 286.0$ | | AIC = 292.0 | | $-2l(\boldsymbol{\theta}) = 280.2$ | | AIC = 288.2 | |

Figure 3 shows that the systematic components for both parameters $\mu$ and $\sigma$ can model the non-proportional hazards well. Figure 4 clearly reveals that we cannot model the non-proportional hazards by taking only the systematic component for $\mu$.

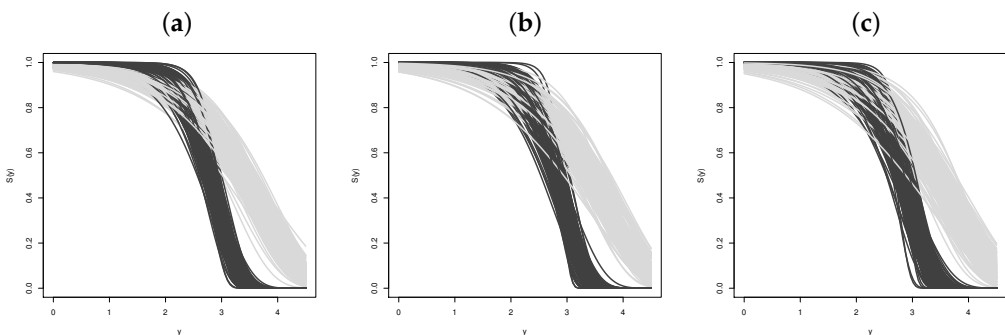

**Figure 3.** Estimated survival function with covariables on the parameters $\mu$ and $\sigma$: (**a**) Interval-censored 100%. (**b**) Interval-censored 90%. (**c**) Interval-censored 70%.

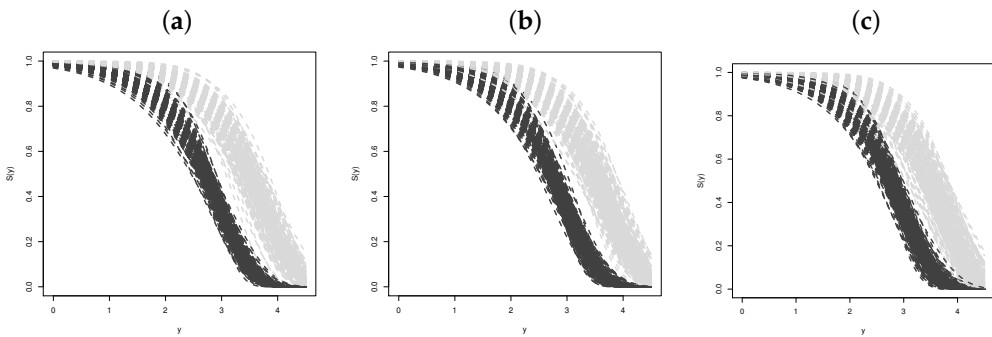

**Figure 4.** Estimated survival function with covariables on the parameter $\mu$: (**a**) Interval-censored 100%. (**b**) Interval-censored 90%. (**c**) Interval-censored 70%.

## 5. Applications

We present two applications in different areas. The first application refers to experiments in veterinary science and the second in medicine.

### 5.1. Regression for the Supplementation Animal Data

The dataset refers to a study of dairy herds developed by the Department of Veterinary Medicine of the Federal University of Lavras. The objective was to verify if the supplementation offered to the herd was influencing the ovulation time of the animals. The experiment was carried out with fifty dairy cows divided into two groups: the control group corresponds to the animals without supplementation and the treatment group corresponds to the animals treated with supplementation to induce ovulation.

The response variable is the time (in days) after delivery until the first ovulation, but only for some animals was it possible to know the exact time of the first ovulation. For

the other animals, the only information is the exact time that failure occurred in a time interval, which characterizes the presence of interval censoring. The following variables are considered (for $i = 1, \ldots, 50$):

- $y_i$: Logarithm time after delivery until the first ovulation;
- $x_{i1}$: Treatment (0 = control, 1 = supplementation);
- $x_{i2}$: Ovary (0 = right, 1 = left);
- $x_{i3}$: Number of pups (0 = two pups, 1 = two more pups).

The dataset is formed by dichotomous covariables. The estimated survival curves in Figure 5 created using the Turnbull method examine the behavior of the covariables in relation to the logs of ovulation times. Figure 5 indicates that the assumption of proportional risks is not satisfied for the current dataset. So, we adopt the regression in Equation (4) to explain these data.

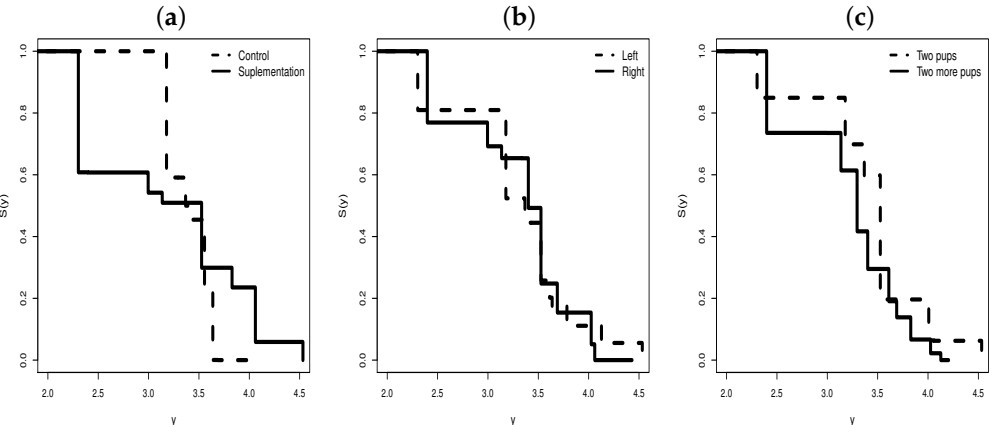

**Figure 5.** Estimated survival curve by the Turbull method for supplementation animal data: (**a**) Treatment ($x_1$). (**b**) Ovary ($x_2$). (**c**) Number of pups ($x_3$).

Consider the log-Weibull regression

$$y_i = \mu_i + \sigma_i\, z_i,$$

for the current data, where $z_i$ has the density function (3) and

$$\mu_i = \beta_{10} + \beta_{11} x_{i1} + \beta_{12} x_{i2} + \beta_{13} x_{i3} \quad \text{and} \quad \sigma_i = \exp(\beta_{20} + \beta_{21} x_{i1} + \beta_{22} x_{i2} + \beta_{23} x_{i3}).$$

The MLEs from the two regressions are reported in Table 3. The $p$-value of 0.003 for the estimate of $\beta_{21}$ indicates a significant difference between the levels of the treatment to explain the variability of the logs of the failure times.

**Table 3.** Some findings from the fitted log-Weibull regression.

| Only One Component for $\mu$ | | | | Two Components for $\mu$ and $\sigma$ | | | |
|---|---|---|---|---|---|---|---|
| Parameter | Estimate | SE | $p$-Value | Parameter | Estimate | SE | $p$-Value |
| $\beta_{10}$ | 3.649 | 0.159 | <0.001 | $\beta_{10}$ | 3.601 | 0.105 | <0.001 |
| $\beta_{11}$ | 0.123 | 0.146 | 0.401 | $\beta_{11}$ | -0.002 | 0.154 | 0.991 |
| $\beta_{12}$ | 0.045 | 0.145 | 0.759 | $\beta_{12}$ | 0.230 | 0.131 | 0.085 |
| $\beta_{13}$ | −0.186 | 0.144 | 0.202 | $\beta_{13}$ | −0.132 | 0.131 | 0.320 |
| $\sigma$ | 0.497 | 0.062 | - | $\beta_{20}$ | −1.304 | 0.275 | <0.001 |
| | | | | $\beta_{21}$ | 0.867 | 0.274 | 0.003 |
| | | | | $\beta_{22}$ | −0.304 | 0.276 | 0.276 |
| | | | | $\beta_{23}$ | 0.293 | 0.313 | 0.353 |
| $-2\,l(\hat{\theta}) = 157.700$ | | AIC = 167.700 | | $-2\,l(\hat{\theta}) = 148.000$ | | AIC = 164.000 | |

Figure 6a displays the random residuals within the interval $(-3, 3)$. The QQ plot with a generated envelope is reported in Figure 6b to verify the response distribution. Both plots support that the two components of the fitted regression are necessary to explain these data.

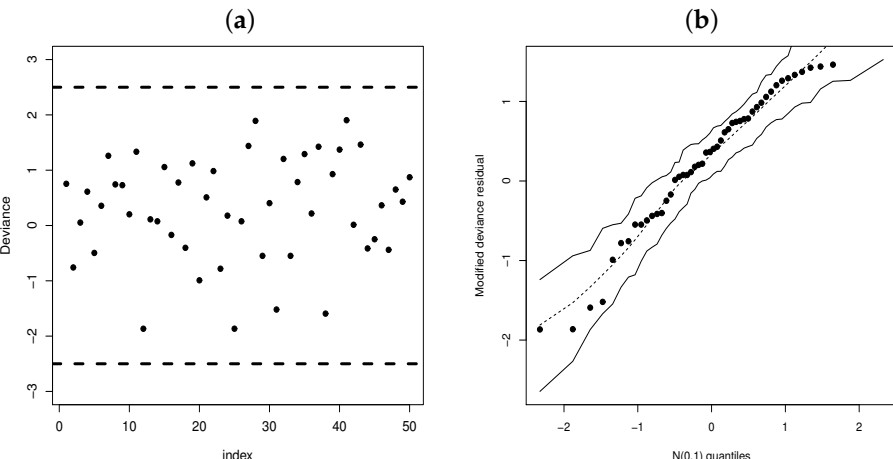

**Figure 6.** (**a**) Index plot of the residuals. (**b**) QQ plot with an envelope.

Plots of the empirical and estimated survival functions and the estimated hazard rates are given in Figures 7 and 8, respectively. We note an increasing curve for the ovulation time data and the non-proportionality of the hazards, which supports the new regression with two components.

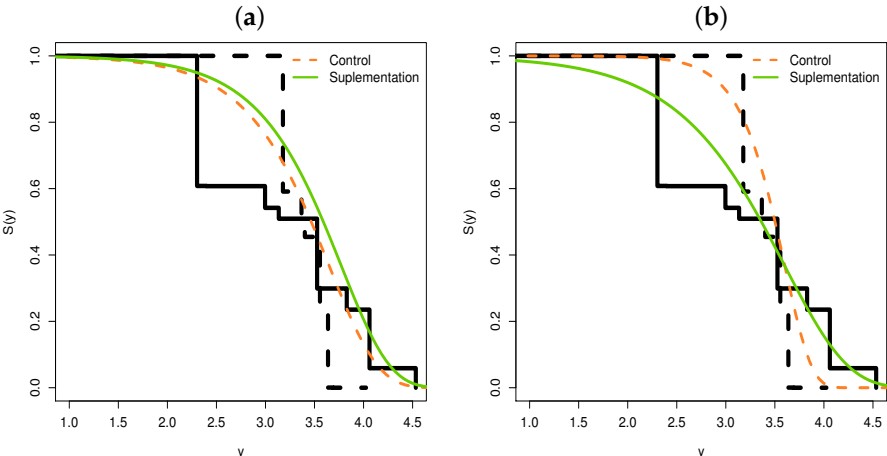

**Figure 7.** Estimated survival curve by the Turbull method for ovulation time data: (**a**) Systematic component for $\mu$. (**b**) Systematic components for $\mu$ and $\sigma$.

**Interpretations**

- The levels of the control and supplementation of the treatment are different, explaining the variability in the log ovulation time.
- From Figure 7b, we note that before $\exp(3.5) = 33$ days (approximately), the treatment control level has a longer survival time than the supplementation level.
- After 33 days, we note the opposite, i.e., the survival time of the supplementation level is longer than the control level in relation to the time of ovulation.
- Thus, if the supplement is applied at longer intervals, the supplement level would have a longer survival time compared with the control.
- We can also note that this change in 33 days is captured by the systematic part of the parameter $\sigma$.

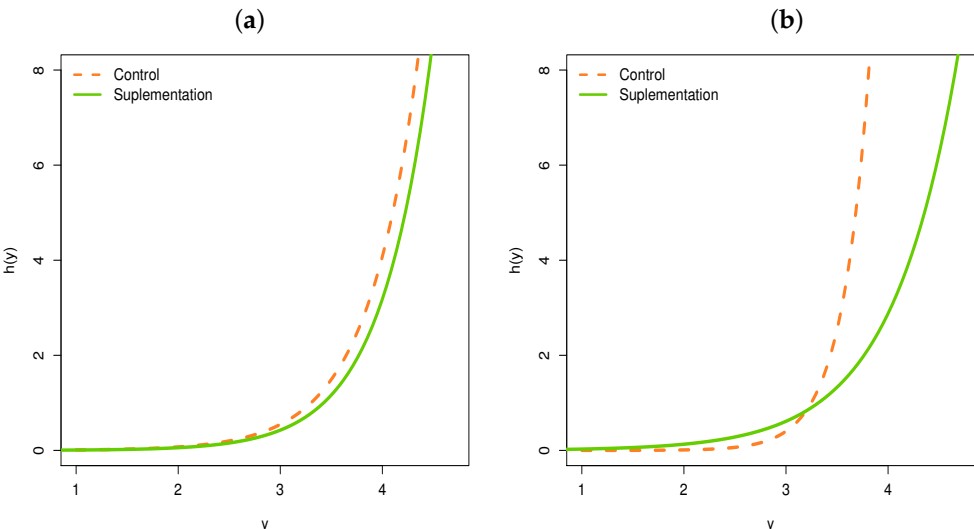

**Figure 8.** Estimated hazard rate curve for ovulation time data: (**a**) Systematic component for $\mu$. (**b**) Systematic components for $\mu$ and $\sigma$.

### 5.2. Regression for Breast Cancer Data

We investigate the log-Weibull regression in the presence of interval-censored data [19] when proportional risks are not satisfied. These data are taken from a retrospective study reported by [20,21] to compare the cosmetic effects of radiotherapy alone versus radiotherapy and adjuvant chemotherapy on women with early breast cancer.

In this study, we consider the following variables (for $i = 1, \ldots, 94$):

- $y_{i1}$: Logarithm of time (in months) to first appearance of moderate or severe breast retraction;
- $x_{i1}$: Type of treatment (0 = radiotherapy and chemotherapy, 1 = radiotherapy).

The dataset is composed of a dichotomous covariable. Figure 9 displays the estimated survival curves obtained using the Turnbull method to verify the behavior of this covariable in relation to the log retraction time.

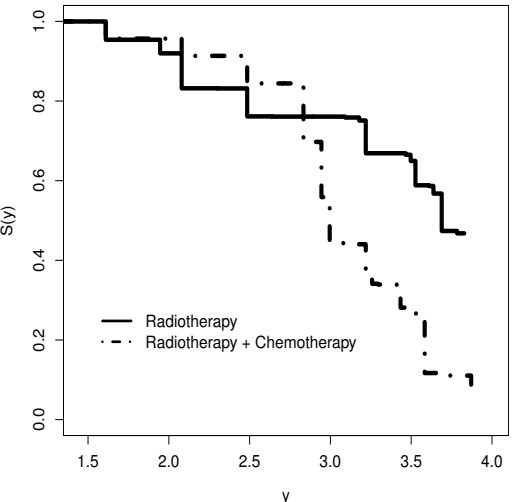

**Figure 9.** Estimated survival curve by the Turbull method for the breast cancer data by treatment: radiotherapy and radiotherapy + chemotherapy.



The proposed regression for the breast cancer data takes the form

$$y_i = \mu_i + \sigma_i z_i,$$

where $z_i$ has the density function (3) and the parameters are

$$\mu_i = \beta_{10} + \beta_{11} x_{i1} \quad \text{and} \quad \sigma_i = \exp(\beta_{20} + \beta_{21} x_{i1}).$$

Table 2 provides some results from the fitted regressions. For a 5% significance level, the retraction time has significantly different effects from radiotherapy and radiotherapy plus chemotherapy, considering both location and dispersion parameters.

Figure 10a gives the plots of the modified deviance residuals against the index. Figure 10b provides the QQ plot and generated envelope. These plots support the wider log-Weibull regression for the current data.

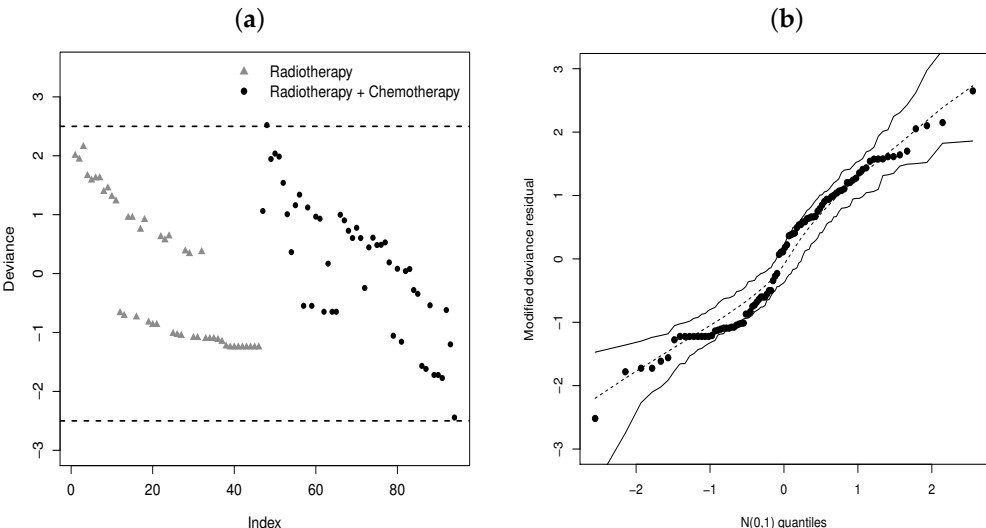

**(a)**  **(b)**

**Figure 10.** (**a**) Index plot of the residuals. (**b**) QQ plot with envelope.

In Figure 11, we present the plots for the empirical and estimated survival functions for the log-Weibull regression. Figure 11a considers only the systematic component for $\mu$, whereas Figure 11b considers the systematic components for $\mu$ and $\sigma$.

The estimated hrf displayed in Figure 12 indicates increasing shapes for the ovulation time. Figure 12a refers to just one component for $\mu$, whereas Figure 12b refers to two components for $\mu$ and $\sigma$. These plots support the non-proportionality of the hrf and a regression with two components for a better fit to these data.

**Interpretation for $\mu$**

- There is a significant difference between the levels of radiotherapy and chemotherapy and radiotherapy in terms of the covariable treatment in relation to the log time of the first appearance of moderate or severe breast retraction.

**Interpretations of $\sigma$**

- There is a significant difference between the levels of radiotherapy and chemotherapy and radiotherapy in terms of the covariable treatment in relation to the variability of the logarithm of the time of the first appearance of moderate or severe breast retraction.
- We note from Figure 11b that before $\exp(2.5) = 12$ months (approximately), the radiotherapy and chemotherapy level of treatment has a longer survival time than the radiotherapy level, but this difference is not significant.
- After 12 months, we note the opposite, i.e., the survival time of the radiotherapy level is longer than that of the radiotherapy and chemotherapy level in relation to the time of the first appearance of moderate or severe breast retraction.

- So, we note that 12 months of applying radiotherapy and chemotherapy to the patient makes them less immune.
- We can also note that this change after 12 months is captured by the systematic part of $\sigma$.

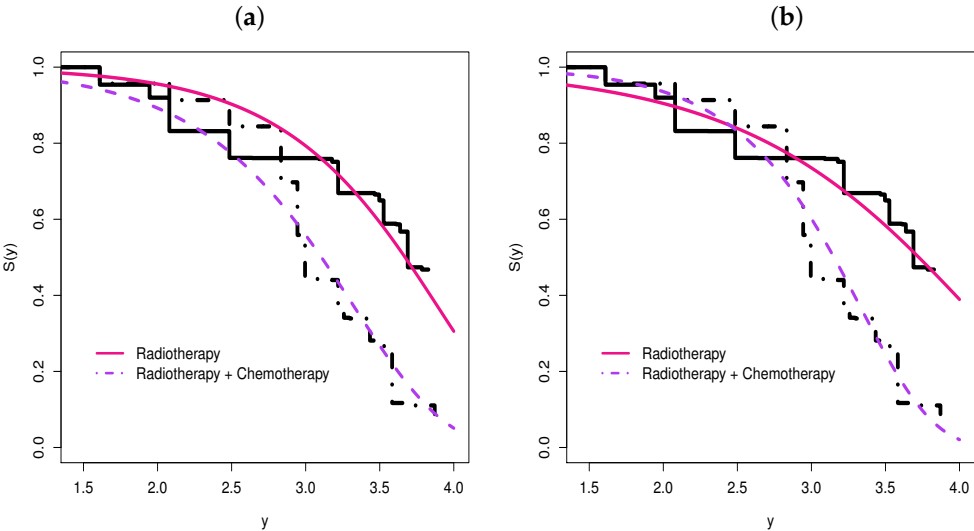

**Figure 11.** Estimated survival curve by the Turbull method for breast cancer: (**a**) One component for $\mu$. (**b**) Two components for $\mu$ and $\sigma$.

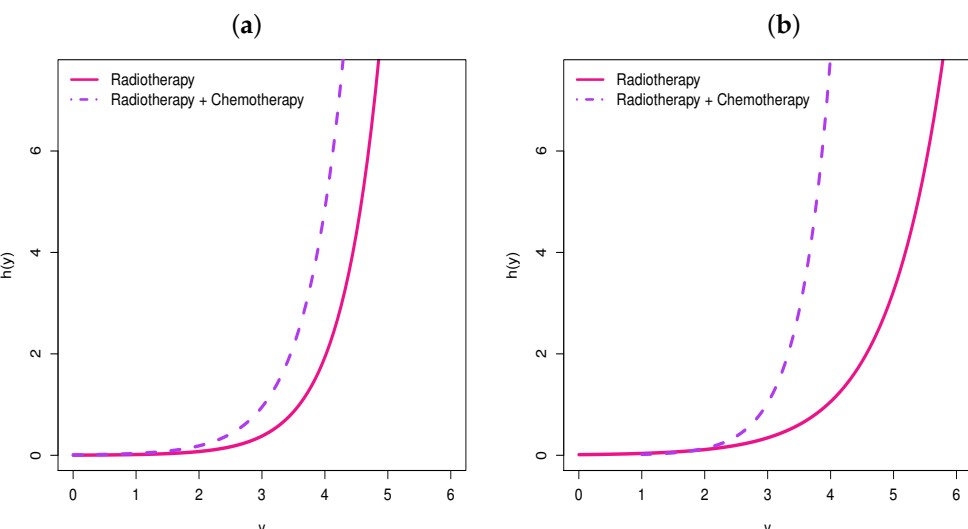

**Figure 12.** Estimated hazard rate curves for breast cancer: (**a**) One component for $\mu$. (**b**) Two components for $\mu$ and $\sigma$.

## 6. Conclusions

We define and study a new log-Weibull regression for interval-censored data with two systematic components for the location and scale parameters, whose risks are not proportional. The parameters are estimated by maximum likelihood and some Monte Carlo simulations are used to investigate the accuracy of the estimates and the normal approximation for the deviance residuals. We show significant differences between two treatments for the supplementation of dairy cows. We emphasize the utility of the log-Weibull regression in two applications to real data. The datasets can be obtained by contacting the main author. Several future works can be considered, based on the assumption of non-proportional hazards; for example, the research by Hashimoto et al. [1] referring to the regression model based on the log-exponentiated Weibull distribution for interval-censored data can be

generalized considering the assumption of non-proportional hazards. Analogously, we can extend the research presented by Hashimoto et al. [22] and, in this case, the extension will be related to regression models with a cure fraction for interval-censored data and non-proportional hazards. The study presented by Yang et al. [23] can be extended to non-proportional hazards models for interval-censored data considering two systematic components. Other future works may include, for example, the use of regression models with random effects for interval-censored data with non-proportional hazards in the form of group structures or correlated data, and, finally, the use of regression models for interval-censored data under the assumption of hazards not being proportional to informative censorship.

**Author Contributions:** All the authors contributed equally to this work. All authors have read and agreed to the published version of the manuscript.

**Funding:** This research received no external funding.

**Institutional Review Board Statement:** Not applicable.

**Informed Consent Statement:** Not applicable.

**Data Availability Statement:** Stated in the text.

**Conflicts of Interest:** The authors declare no conflict of interest.

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
