# Peer review of "Interval-Censored Regression with Non-Proportional Hazards with Applications"

_stats, doi:10.3390/stats6020041_

Round 1
Reviewer 1 Report
The problem is well-laid out. However, the paper is written on a purely heuristic level without containing hard statement. Weibull AFT models are common in handling survival times! Since the paper proposes no methodological advance, my view is that extensive simulations ought to be included to deserve publication. The authors may consider simulations that are carried out for finite sample sizes (less than 50) and under heavy censoring (larger than 50%), if possible. By looking at the results under this set-up, it may be possible to evaluate that the approach in the paper actually leads to good performance in practice.
The paper was written in a big-hurry. An example includes “verify the fir of the model.” The reader advise the authors to proofread the paper.
Author Response
See attached letter of reply.

Reviewer 2 Report
Please find attached.

English language is fine to me.
Author Response
See attached letter of reply.

Round 2
Reviewer 1 Report
The contents of the revised manuscript are improved on the previous version. I am pleased that the authors have carefully addressed and managed the issues raised by the reviewers.
Reviewer 2 Report
I have no further comments.